# Non-Similar Solution of G-jitter Induced Unsteady Magnetohydrodynamic Radiative Slip Flow of Nanofluid

**M.J. Uddin [1,2], W.A. Khan [3,*], O. Anwar Bég [4] and A. I. M. Ismail [2]**

[1]   American International Univerity-Bangladesh, Kuril, Dhaka 1229, Bangladesh; drjashim@aiub.edu
[2]   School of Mathematical Sciences, Universiti Sains Malaysia, Penang 11800, Malaysia; izani@cs.usm.my
[3]   Department of Mechanical Engineering, College of Engineering, Prince Mohammad Bin Fahd University, Al Khobar 31952, Saudi-Arabia
[4]   Fluid Mechanics, Bio-Propulsion and Nano-systems, Aeronautical and Mechanical Engineering, Salford University, Manchester, UK M54WT, USA; O.A.Beg@salford.ac.uk
*    Correspondence: wkhan1956@gmail.com

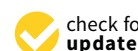

**Featured Application: The results reported have many engineering applications, such as electromagnetic micro-pumps and nanomechanics in microgravity environments.**

**Abstract:** We present a mathematical model and numerical simulation of the unsteady 2-D g-jitter-free and forced the convective flow of water-based nanofluid from a flat plate, considering both the velocity slip and thermal slip conditions imposed on the wall of the plate. The Darcian model is used, and both cases of a calm and moving free stream are considered. In place of the extensively used linearly varying radiative heat flux, the nonlinearly varying heat flux calculation is applied to produce practically useful results. Further, we incorporate the "zero mass flux boundary condition" which is believed to be more realistic than the earlier extensively used "actively" controlled model. The parameter influences the non-dimensional velocity, temperature, nanoparticle volume fraction, skin friction and heat transfer rates are visualized graphically and discussed in detail. Special cases of the results are benchmarked with those existing in the literature, and a good arrangement is obtained. It is found that the rate of heat transfer is lower for the calm free stream rather than the moving free stream.

**Keywords:** g-jitter; radiation; zero mass flux; slip flow; nanofluid; materials processing

## 1. Introduction

G-jitter induced convection occurs due to forces associated with buoyancy, which is produced by the interaction of temperature and nanoparticle concentration differences and a gravitational field. G-jitter induced convection has many applications. Examples include motors, pumps, spacecraft maneuvers, and crew motions. Interest is growing, due to g-jitter induced streaming formed in microgravity environments. The residual acceleration associated with space vehicles is known to produce buoyancy-driven convection. A good number of authors investigated g-jitter induced flow with heat/mass transfer, due to its many engineering applications. For example, Amin [1] examined heat transfer, and fluid flow in the vicinity of a sphere emerged in a zero-gravity environment. Time-dependent flows in a float zone induced by g-jitter were explored by Chen et al. [2]. Li [3] addressed the g-jitter driven flows in the presence of a magnetic field (for a system which used two parallel plates having unlike temperatures) observing that the g-jitter frequency, magnetic field and temperature difference all affected the flow field and heat transfer characteristics. Rees and

Pop [4] investigated the modifications in flow phenomena near the surface of the wall established because of a constant temperature by incorporating time-periodic variations in the acceleration, due to gravity. In another paper, Rees and Pop [5] investigated the influences of g-jitter on the flow in the neighborhood of the forward stagnation point of a two-dimensional symmetric body. They concluded that the emerging parameters had a significant effect on the friction, as well as heat transfer rates. Later, Rees and Pop [6] scrutinized the flow at the vicinity of the boundary made by a constant temperature by incorporating time-periodic variations in the acceleration, due to gravity. Multi-diffusive convective flow driven by g-jitter in a microgravity environment was studied by Shu et al. [7]. They performed extensive simulations for flow, thermal and mass transports, concluding that the local concentration gradient remained approximately constant in time and that an increase in g-jitter force promotes nonlinear convective effects. In another study, Shu et al. [8] investigated g-jitter related double-diffusive flows, temperature and concentration distributions with binary alloy melt systems and magnetic field, concluding that the main melt flow followed roughly a linear superposition of velocity components and was induced by individual g-jitter components, irrespective of whether a magnetic field was present or absent. They further identified that the magnetic damping effect was more pronounced on the velocity associated with the most significant g-jitter component or the g-jitter spiking peaks present. G-jitter effects on the flow from a sphere were investigated by Amin et al. [9]. They presented analytical results for different cases using the matched asymptotic expansions method, demonstrating that the flow and thermal behavior follow intricate trends based on the buoyancy, Prandtl and Schmidt numbers. Using numerical techniques, Sharidan et al. [10] obtained a solution for natural convective of a micropolar fluid at the forward stagnation point of a two-dimensional symmetric body, taking into account both the cases when the spin gradient on the wall are zero and non-zero, and showing that the emerging parameters significantly influence the flow characteristics. Pop et al. [11] studied natural convective flow near a three-dimensional stagnation flow with a step-change in surface conditions and computed the influences of a small change of gravitational field. The effects of g-jitter on the buoyancy-induced flow in a binary viscoelastic fluid-saturated porous medium were described by Suthar et al. [12]. They used the Rivlin-Ericksen model to investigate the viscoelasticity of the fluid and examined the stability of the system using Floquet theory. Their results showed that g-jitter had a vital role in the onset of thermo-solutal convective flow. Rawi et al. [13] studied the impact of g-jitter on micropolar convective flow from an extendable sheet. They found that the rate of heat and mass transfer increased, whereas, the reduced skin friction decreased as the angle of inclination increased. It was further noted that as the buoyancy ratio parameter increased, the rate of heat and mass transfer, as well as reduced skin friction, were decreased. Afiqah et al. [14] studied the combined convective flow of a second-grade non-Newtonian fluid from a tilted stretching surface.

Many manufacturing processes involve both high-temperature effects and porous media (Yih [15], Zhang et al. [16]). Radiation heat transfer is significant in the former case (Mahapatra et al. [17]). In many technological applications, such as hypersonic flight, missile reentry, rocket combustion chambers, solar pond, fluidized bed heat exchangers, and gas-cooled nuclear reactors, etc., *radiation* as a method of energy transmission also plays a significant role (Ozisik [18]). Porous media arise in petroleum engineering, foam manufacture, geothermics, among other areas (Dehghan et al. [19]). Porous media are also deployed in high-temperature heat exchangers, combustion devices, filtration systems, etc. Porous materials are cooled by forcing the liquid/gas through solid capillaries and are also used to insulate a heated body to maintain its temperature (Hernández and Zueco [20]). Further implementations of porous media in 21st-century technologies include solar collectors to increase efficiency for storage (Dehghan et al. [19]). Numerous studies of the radiative-convective flows (which focus on illustrating the radiative heat transfer contribution in coupled processes), in porous media, have been reported, and are motivated by among other areas, thermal insulation engineering and materials fabrication systems. Some relevant recent studies include Vafai and Tien [21], Takhar et al. [22], and, recently, Motsumi and Makinde [23]. These investigations have generally employed algebraic flux models to simulate the uni-directional radiative contribution to heat transfer and include the Rosseland diffusion,

Cogley-Vincenti-Giles non-gray flux model, Schuster-Schwartzchild two-flux model, Chandrasekhar discrete ordinates model, etc. This circumvents the need to solve the formidable integrodifferential radiative transfer equation and greatly simplifies numerical computations. The Rosseland model is the most amenable in boundary layer flows and permits radiative effects to be considered using a unique dimension-free parameter (Rosseland number). Many authors [16–23] have applied a linear Rosseland approximation in steady/unsteady flows. However, this model has limited accuracy in high-temperature circumstances, as elucidated in [24] (and the references quoted therein) and leads to inaccuracies even for optically thick scenarios. Several researchers have, therefore, considered the consequence of nonlinear radiation on the flow field and energy transfer, and relevant work in this regard includes Pantokratoras [24], Pantokratoras and Fang [25], Cortell [26], Uddin et al. [27] and Siqqiqa et al. [28]. In the case of fluid-saturated porous media, the Darcy drag force model, is the most popular, although it is only applicable to low-speed viscous-dominated ("creeping") flows. Many scholars (Nield and Kuznetsov [29], Uddin et al. [30], Hady et al. [31], Mahdi et al. [32]) have used this model for porous media nanofluid modeling. Most of these nanofluid studies applied convectional *no-slip* conditions at the solid-fluid interface. However, as mentioned by Wang [33], in many cases, namely, rarefied gas dynamics [34], flow overcoated/rough/striated surfaces [35], and most superhydrophobic nano-surfaces [36], slip on a solid surface takes place. In such situations, the conventional no-slip condition must be replaced with slip boundary conditions of the form, $\overline{u} = N_1 \tau_0$, where $\overline{u}$ is the tangential velocity, $\tau_0$ is the tangential shear stress and $N_1$ is the dimensional slip factor (Navier [37]), $T = D_1 \frac{\partial T}{\partial y}$ where $D_1$ is the thermal slip factor. Behzad et al. [38] investigated the slip effect on a micro-spherical particle. Usually, velocity slip on the wall enhances convection along the surface, while the temperature jump decreases the heat transfer. Hence, neglecting temperature jump resulted in an overestimation of the heat transfer coefficient.

Inspection of the literature indicates that the only available study related to the g-jitter mixed convective flow of nanofluids has been presented by Uddin et al. [39] who noted that there was a sinusoidal nature exhibited by the Nusselt numbers for both Cu-water and $Al_2O_3$-water nanofluids. This study, however, assumed the nanofluid to be electrically non-conducting. It transpires that magnetohydrodynamic (MHD) mixed convective g-jitter flow in a Darcian porous medium with passively controlled boundary conditions and nonlinear radiation effects has, thus far, not been investigated, and this is the aim of the present analysis. This paper is, therefore, a generalization of the recent work of Uddin et al. [39] on Buongiorno-Darcy porous medium g-jitter convection and incorporates nonlinear radiation, magnetic field effects, and zero mass flux boundary conditions. The governing transport equations are reduced to non-similar equations using appropriate transformations. Simulations are then performed using an implicit finite difference method. The computations are validated against special cases with earlier published results. The influence of the emerging parameters on the dimensionless velocity, temperature, nanoparticle volume fraction, friction factor and heat transfer rates are demonstrated clearly with the help of graphs and tables. The analysis is relevant to magnetic nanofluid materials processing transport phenomena.

## 2. Description and Formulation of the Governing Equations

Consider two-dimensional boundary layer flow of a viscous, non-compressible nanofluid over a nonlinearly radiating plate adjacent to a Darcian porous medium. Free stream velocity is defined by $u_e = Kax$. Here $a$ is constant having dimension 1/s, $K$ is a constant which relates to fraction of stationary versus moving free stream. The plate surface is exposed to a momentum slip and thermal slip, in addition to a zero mass-flux boundary condition.

Figure 1 displays the schematic of the problem under consideration. An exterior magnetic field having uniform strength $B_0$ is applied transverse to the direction of the plate. We ignore the induced magnetic field, due to a sufficiently low value of magnetic Reynolds number. The *x*-axis is aligned with the plate, while the *y*-axis is perpendicular to the plate. The gravitational acceleration is represented by expression $\mathbf{g}^*(t) = g_0[1 + \varepsilon \cos(\pi \omega t)]\vec{K}$ (]) (Sharidan et al. [10]). Here $g_0$ is the time-averaged value

of the acceleration, due to gravity. $\mathbf{g}^*(t)$ is orientated to the direction of the unit vector $\overrightarrow{K}$ (concerned within the upward trend) $\varepsilon$ is a scaling parameter, $t$ is the time, and $\omega$ represents oscillation related to frequency (Sharidan et al. [10]). The forcing can be regarded as *a* perturbation of the mean gravity when $\varepsilon << 1$. Three different boundary layers (velocity, thermal, and concentration) are formed in the vicinity of the surface of the plate. The flow is driven by both the thermal and nanoparticle species buoyancy forces. It is assumed that the velocities are adequately low; hence, the Darcy drag force model is valid. We adopt Darcy's law for an isotropic, homogenous porous medium with constant permeability $k_p$ (m$^2$). The Boussinesq approximation is used. Based on the above simplifications, following Buongiorno [40], and using the order of magnitude analysis (OMA), the transient balance equations are presented (in a dimensional form), respectively, as follows (Sharidan et al. [10]):

$$\frac{\partial u}{\partial x} + \frac{\partial v}{\partial y} = 0,\tag{1}$$

$$\begin{aligned}\frac{\partial u}{\partial t} + u\frac{\partial u}{\partial x} + v\frac{\partial u}{\partial y} = Ku_e\frac{du_e}{dx} + \frac{\mu}{\rho}\frac{\partial^2 u}{\partial y^2} - \frac{\mu}{k_p\rho}(u - Ku_e) - \frac{\sigma B_0^2}{\rho}(u - Ku_e)\\ + [\mathbf{g}*(t)\beta_T(T - T_\infty) + \mathbf{g}*(t)\beta_C(C - C_\infty)]\left(\frac{x}{L}\right),\end{aligned}\tag{2}$$

$$\frac{\partial T}{\partial t} + u\frac{\partial T}{\partial x} + v\frac{\partial T}{\partial y} = \alpha\frac{\partial^2 T}{\partial y^2} + \tau_1\left[D_B\frac{\partial\phi}{\partial y}\frac{\partial T}{\partial y} + \frac{D_T}{T_\infty}\left(\frac{\partial T}{\partial y}\right)^2\right] - \frac{1}{\rho_f c_p}\frac{\partial q_r}{\partial y},\tag{3}$$

$$\frac{\partial C}{\partial t} + u\frac{\partial C}{\partial x} + v\frac{\partial C}{\partial y} = D_B\frac{\partial^2 C}{\partial y^2} + \frac{D_T}{T_\infty}\frac{\partial^2 T}{\partial y^2}.\tag{4}$$

Following Wang [33], Uddin et al. [41] and Kuznetsov and Nield [42], and based on the inherent assumptions outlined earlier, the relevant initial, wall and far-field conditions can be written as:

$$t < 0,\, u = v = 0 \quad \text{any } x,\, y$$

$$\begin{aligned}&t > 0,\, u = u_{\text{slip}},\, v = 0,\, T = T_w + T_{\text{slip}},\, D_B\frac{\partial C}{\partial y} + \frac{D_T}{T_\infty}\frac{\partial T}{\partial y} = 0 \text{ at } y = 0,\\ &u \to u_e,\, T \to T_\infty,\, C \to C_\infty \text{ as } y \to \infty,\end{aligned}\tag{5}$$

Here $\alpha = \frac{k}{(\rho c)_f}$: Thermal diffusivity, $\tau_1 = \frac{(\rho c)_p}{(\rho c)_f}$: Ratio of heat capacity of the nanoparticle and fluid, $(u, v)$: Components of velocity, $u_{\text{slip}} = \frac{N_1\mu}{\rho}\frac{\partial u}{\partial y}$ velocity slip with velocity slip factor $N_1$, $T_{\text{slip}} = D_1\frac{\partial T}{\partial y}$: Thermal slip with thermal slip factor $D_1$, $\mu$ kinematic viscosity of the fluid, $\rho$: Density of the fluid, $D_B$: Diffusion coefficient for Brownian motion, $D_T$: Diffusion coefficient for thermophoresis, $(\beta_T, \beta_C)$ coefficients of thermal and mass expansions, $L = \sqrt{\frac{\alpha}{a}}$: Characteristic length. Note that $K = 0$ implies flow without a pressure force, and $K = 1$ corresponds to flow, due to pressure force. It is assumed that the boundary layer is optically thick, and the nonlinear Rosseland diffusion approximation for radiation is valid. The radiative heat flux (with significant absorption), as reported by Sparrow and Cess [43], is defined as $q_r = -\frac{4\sigma_1}{3k_1}\frac{\partial T^4}{\partial y}$, where $\sigma_1$ ($= 5.67 \times 10^{-8}$ W/m$^2$-K$^4$) is the Stefan-Boltzmann constant and $k_1$ (m$^{-1}$) is the Rosseland mean absorption coefficient. To reduce the number of variables and equations, we invoke the dimensional stream function $\psi$ which satisfies the equations $u = \frac{\partial\psi}{\partial y}$, $v = -\frac{\partial\psi}{\partial x}$. Note that $\psi$ satisfies the continuity Equation (1) automatically. Let us invoke the following co-ordinate transformations (Uddin et al. [39], Yih et al. [44]):

$$\begin{aligned}&\tau = \omega t,\, \eta = \sqrt{\frac{a}{\alpha}}y,\, u = ax\frac{\partial f(\eta,\tau)}{\partial\eta},\, v = -\sqrt{a\alpha}f(\eta,\tau),\\ &\theta = \frac{T - T_\infty}{T_w - T_\infty} = \theta(\eta,\tau),\, \phi = \frac{C - C_\infty}{C_w - C_\infty} = \phi(\eta,\tau),\, g(\tau) = \frac{\mathbf{g}^*(t)}{g_0},\end{aligned}\tag{6}$$

Implementation of Equation (6) in Equations (2)–(4) yields:

$$\begin{aligned}\text{Pr}\frac{\partial^3 f}{\partial\eta^3} + f\frac{\partial^2 f}{\partial\eta^2} - \left(\frac{\partial f}{\partial\eta}\right)^2 + (1 + \varepsilon\cos\pi\tau)\lambda[\theta + Nr\phi] - \left(\frac{1}{Da} + M^2\right)\left(\frac{\partial f}{\partial\eta} - K\right)\\ + K = \Omega\frac{\partial^2 f}{\partial\tau\partial\eta},\end{aligned}\tag{7}$$

$$\frac{\partial^2 \theta}{\partial \eta^2} + f\frac{\partial \theta}{\partial \eta} + Nb\frac{\partial \theta}{\partial \eta}\frac{\partial \phi}{\partial \eta} + Nt\left(\frac{\partial \theta}{\partial \eta}\right)^2 + \frac{4}{3R}\frac{\partial}{\partial \eta}\left[\{1 + (T_r - 1)\theta\}^3 \frac{\partial \theta}{\partial \eta}\right] = \Omega\frac{\partial \theta}{\partial \tau}, \tag{8}$$

$$\frac{\partial^2 \phi}{\partial \eta^2} + Le\, f\frac{\partial \phi}{\partial \eta} + \frac{Nb}{Nt}\frac{\partial^2 \theta}{\partial \eta^2} = Le\,\Omega\frac{\partial \phi}{\partial \tau}. \tag{9}$$

with boundary conditions:

$$\frac{\partial f}{\partial \eta}(\tau,0) = \delta_v\frac{\partial^2 f}{\partial \eta^2}(\tau,0), f(\tau,0) = 0,\ \theta(\tau,0) = 1 + \delta_T\frac{\partial \theta}{\partial \eta}(\tau,0),$$
$$Nb\phi'(0) + Nt\theta'(0) = 0, \frac{\partial f}{\partial \eta}(\tau,\infty) = K, \theta(\tau,\infty) = \phi(\tau,\infty) = 0. \tag{10}$$

The parameters in the transformed Equations (7)–(10) take the following definitions: $\Pr = \frac{\nu}{\alpha}$ (Prandtl number), $\Omega = \frac{\omega}{a}$ (frequency), $\varepsilon$ (amplitude of modulation), $\lambda = \frac{g_0\beta_T(T_w - T_\infty)}{a^2 L}$ (mixed convection parameter), $Nr = \frac{\beta_C\Delta C}{\beta_T\Delta T}$ (buoyancy ratio), $Da = \frac{ak_p}{\nu}$ (modified Darcy number), $Nt = \frac{\tau_1 D_T(T_w - T_\infty)}{T_\infty\alpha}$ (thermophoresis parameter), $Nb = \frac{\tau_1 D_B(C_w - C_\infty)}{\alpha}$ (Brownian motion), $Le = \frac{\nu}{D_B}$ (Lewis number), $T_r = \frac{T_w}{T_\infty}$ (wall temperature excess ratio), $R = \frac{kk_1}{4\sigma_1 T_\infty^3}$ (Rosseland conduction-radiation parameter), $M^2 = \frac{\sigma B_0^2}{a\rho}$ (magnetic field parameter), $\delta_v = \sqrt{\frac{a}{\alpha}}\frac{\mu N_1}{\rho}$ (velocity slip), $\delta_T = \sqrt{\frac{a}{\alpha}}D_1$ (thermal slip).

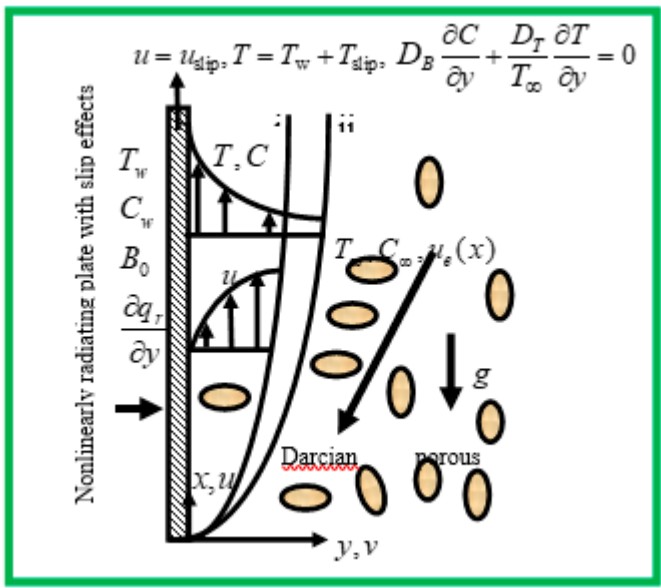

**Figure 1.** Flow model and physical coordinate system.

## 3. Physical Quantities

The performance of numerous devices, such as microfluidic/nanofluidic pumps and sensors, thermal management and micro-gravity spacecraft laboratory components, depends strongly on key wall gradients of the primary variables (velocity, temperature, etc.), i.e., surface shear stress (friction factor), wall heat and mass transfer rates. The information related to wall property variation provides data that may bring about an enhancement in the design of such devices for augmented performance and efficiency (Siddiqa et al. [28]). Thus, skin friction and heat transfer rate are vital quantities and are given by:

$$C_{fx} = \frac{\tau_w}{\rho u_w^2},\ Nu_x = \frac{xq_w}{k(T_w - T_\infty)}, \tag{11}$$

where $\tau_w, q_w$, are shear stress, wall heat flux and are defined as:

$$\tau_w = -\mu\left(\frac{\partial u}{\partial y}\right)_{y=0},\ q_w = -k\left(\frac{\partial T}{\partial y}\right)_{y=0}. \tag{12}$$

Using Equations (6) and (12), we have from Equation (11):

$$\sqrt{\frac{Re_x}{Pr}}C_{fx} = \frac{\partial^2 f}{\partial \eta^2}(\tau,0), \ Pe_x^{-1/2}Nu_x = -\left[1 + \frac{4}{3R}\{1 + (T_r - 1)\theta(\tau,0)\}^3\right]\theta'(\tau,0) \tag{13}$$

where $Re_x = \frac{\rho u_e x}{\mu}$ is the local Reynolds number and $Pe_x = \frac{u_e x}{\alpha}$ is the local Péclet number.

## 4. Computational Solution of Nonlinear Boundary Value Problem

The numerical method employed exploits the Bellman-Kalaba quasilinearization approach, which is documented extensively in the literature [45]. Bellman-Kalaba quasilinearization is a generalized version of the Newton-Raphson method, introduced by the famous American mathematicians, Richard Bellman and Richard Kalaba in the mid-1960s. The technique is ideal for multiple-order, multi-degree nonlinear differential equation systems as encountered in nanofluid dynamics. The method is quadratically convergent, and commences from an initial guess value. The validity of the solution is maintained for a large number of parameter values. It achieves convergence at very high speed, irrespective of the full specification of boundary conditions at the start of the flow domain. The governing differential equations are reduced from a group of multi-order equations to a system of first-order differential equations via a series of substitutions. For a system with $i$ variables, e.g., velocity, temperature, nanoparticle concentration, etc., the following robust algorithm is applied where $j, k$ correspond to different station locations:

$$\frac{\partial g_i^{(k+1)}}{\partial t} = g_i(x_j^k, t) + \sum_{j=1}^{n}\left[\frac{\partial g_i}{\partial x_j}\right]^k (x_j^{k+1} - x_j^k) \tag{14}$$

Here $g_i$ is a general variable, and $t$ is the temporal coordinate (time), $x$ is a general coordinate ($x$, $y$, or $z$) variable and $n$ is the number of steps utilized. A modern perspective on simulation of multi-physical thermal convection flows for two-parameter (spatial-temporal) partial differential equation systems using this method is provided by Bég et al. [46]. The algebraic details are omitted here for brevity. In the absence of the nanoparticle volume fraction equation and additionally setting $\delta_v = \delta_T = \Omega = \lambda = 0$ and $R \rightarrow \infty$ we retrieve the same equations as derived by Yih [44] for an isothermal plate. We benchmark the present computational solutions with those obtained for the case considered by Yih et al. [44], which has also been examined by Sparrow [47], Ariel [48], and Lin [49] to verify the correctness. The results are compared in Tables 1–3, and excellent agreement is achieved. Confidence in the present code is, therefore, justifiably high.

**Table 1.** Comparison of results of skin friction when $Da \rightarrow \infty, \lambda = \Omega = 0, K = Pr = 1$.

| *M* | Sparrow [47] | Ariel [48] | Yih [44] | Present Results |
|---|---|---|---|---|
| 0 | 1.231 | 1.233 | 1.233 | 1.233 |
| 1 | 1.584 | 1.585 | 1.585 | 1.585 |
| 2 | 2.345 | 2.347 | 2.346 | 2.347 |
| 5 | - | 5.148 | 5.148 | 5.148 |
| 10 | - | 10.075 | 10.075 | 10.075 |

**Table 2.** Comparison of results of dimensionless heat transfer when $Nb = Nt = 0, \delta_v = \delta_T = 0$, $Da \rightarrow \infty, R \rightarrow \infty, \lambda = \Omega = M = 0, K = 1$.

| Pr | Sparrow [47] | Lin [49] | Yih [44] | Present |
|---|---|---|---|---|
| 0.001 | - | 0.025 | 0.024829 | 0.025 |
| 0.01 | 0.076 | 0.076 | 0.075973 | 0.076 |
| 0.1 | 0.219 | 0.2196 | 0.219503 | 0.219 |
| 1 | 0.570 | 0.5706 | 0.570465 | 0.570 |
| 10 | 1.349 | 1.339 | 1.338796 | 1.338 |
| 100 | - | 2.986 | 2.986329 | 2.975 |
| 1000 | - | 6.529 | 6.529137 | 6.519 |
| 10,000 | - | 14.158 | 14.158 | 14.142 |

**Table 3.** Comparison of results of dimensionless heat transfer when $Nb = Nt = 0, \delta_v = \delta_T = 0,$ $Da \to \infty, R \to \infty, \lambda = \Omega = 0, K = 1$.

| Pr | M | Yih [44] | Sparrow [47] | Present |
|----|---|----------|--------------|---------|
|  | 0 | 0.219 | 0.219 | 0.219 |
| 0.1 | 1 | 0.224 | 0.224 | 0.224 |
|  | 2 | 0.231 | 0.231 | 0.231 |
|  | 0 | 0.570 | 0.570 | 0.570 |
| 1 | 1 | 0.595 | 0.595 | 0.595 |
|  | 2 | 0.634 | 0.634 | 0.637 |
|  | 0 | 1.339 | 1.349 | 1.338 |
| 10 | 1 | 1.425 | 1.433 | 1.424 |
|  | 2 | 1.567 | 1.574 | 1.566 |

## 5. Illustration of the Results and Interpretation

The transformed Equations (7)–(9) with Equation (10) have been solved using a finite difference method with Bellman-Kalaba quasilinearization, for various values of the involved parameters. The simulated results are plotted in Figures 2–13.

The influence of *Nr*, and *K* on velocity profiles is depicted in Figure 2a,b. It is clear from Figure 2a that velocity decreases as the buoyancy ratio rises both with and without pressure force term. Increasing mixed convection parameter leads to acceleration in the flow (increase in velocity) both with and without buoyancy forces.

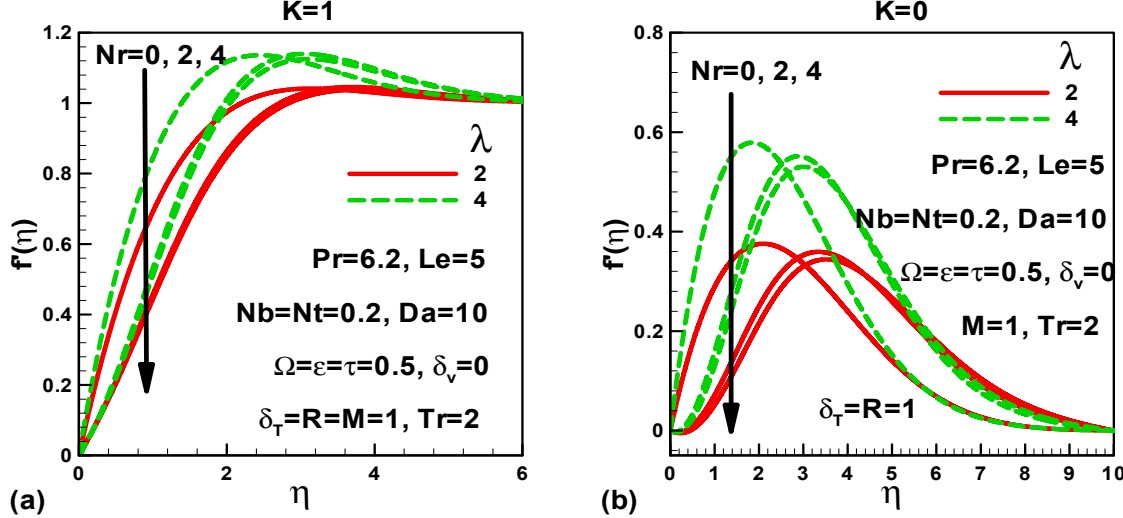

**Figure 2.** Effects of buoyancy ratio and mixed convection parameters on dimensionless velocity (**a**) with pressure gradient and (**b**) without pressure gradient.

Figure 3a,b show the influences of velocity (momentum) slip, as well as thermal slip on the dimensionless velocity profiles. The dimensionless velocity decreases (the momentum boundary layer is thickened) as the thermal slip parameter rises. The velocity slip parameter, however, leads to an augmentation of the velocity in the presence of a pressure force. The influence of the Darcy number and dimensionless time on the velocity is demonstrated in Figure 4a. Velocity falls as the Darcy number rises for both steady-state and unsteady-state flows. The modification in response is due to the pressure gradient parameter, *K*, which counteracts the usual accelerating flow accompanying a boost in Darcy number (rise in permeability, $k_p$) via the corresponding decrease in Darcian body force. The effects of the magnetic field and the Rosseland conduction-radiation parameter are displayed in Figure 4b. Increasing thermal radiation (lower *R* values) lessens the velocity for both hydrodynamic (*M* = 0)

and magnetohydrodynamic ($M \neq 0$) flow. It is further noticed that greater magnetic field provides an enhancement of velocity, and this reverses the conventional Lorentz hydromagnetic drag force effect, due to the existence of the free stream which effectively forces the magnetic field to accelerate the flow [50].

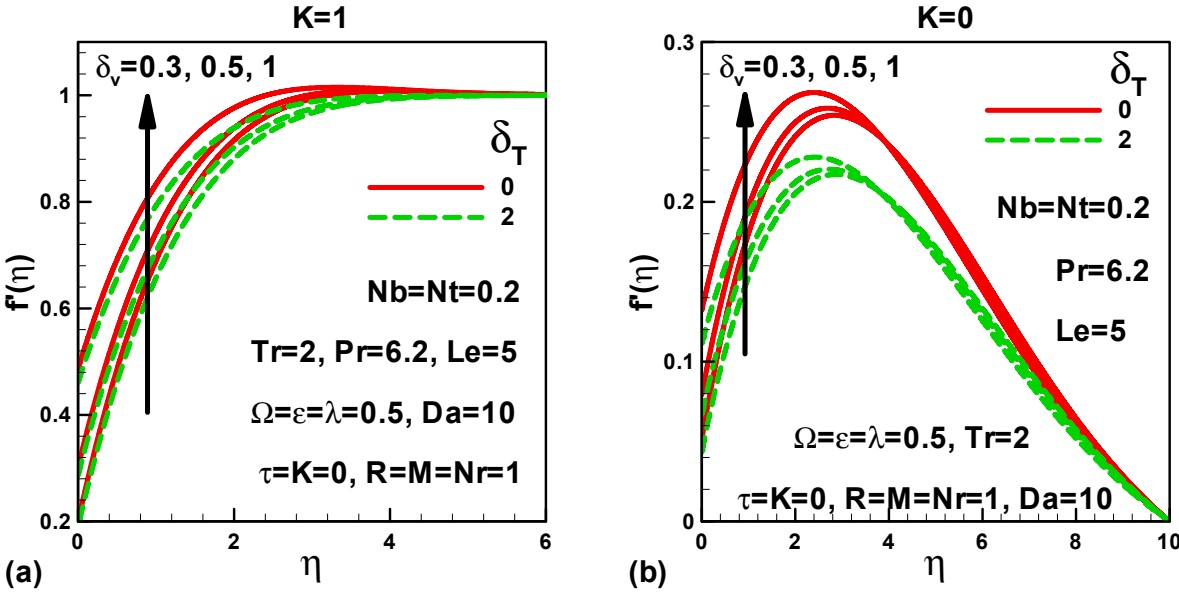

**Figure 3.** Effects of slip parameters on dimensionless velocity (**a**) with pressure gradient and (**b**) without pressure gradient.

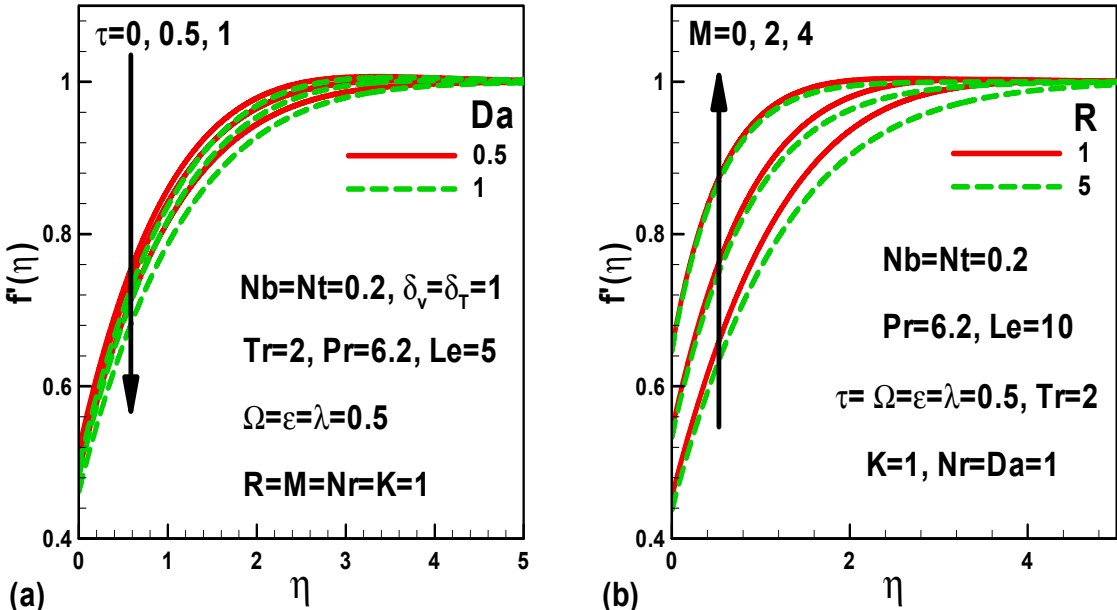

**Figure 4.** Variation of dimensionless velocity with (**a**) Darcy number and dimensionless time and (**b**) with magnetic and radiation parameters.

Figure 5a,b are drawn to display the influences of buoyancy ratio (*Nr*), mixed convective parameter, and pressure gradient parameter (*K*) on temperature profiles. Temperature is enhanced with elevation in buoyancy ratio both with and without pressure gradient (Figure 5a) force. A rise in mixed convective parameter reduces the flow velocity both with and without buoyancy forces. The impacts of velocity slip and thermal slip on the temperature profiles are exhibited in Figure 6a,b. It is found that temperature is reduced as the thermal slip parameter rises, for both slip flow and conventional no-slip flow. Physically

this is because as thermal slip parameter $\delta_T$ rises, the fluid at the vicinity of the plate will not be able to sense the influence of heating of the plate, and reduced heat will be released from the plate to the surrounding fluid. Elevation in velocity slip parameter induces an enhancement in the velocity for both the isothermal plate and plate with thermal slip.

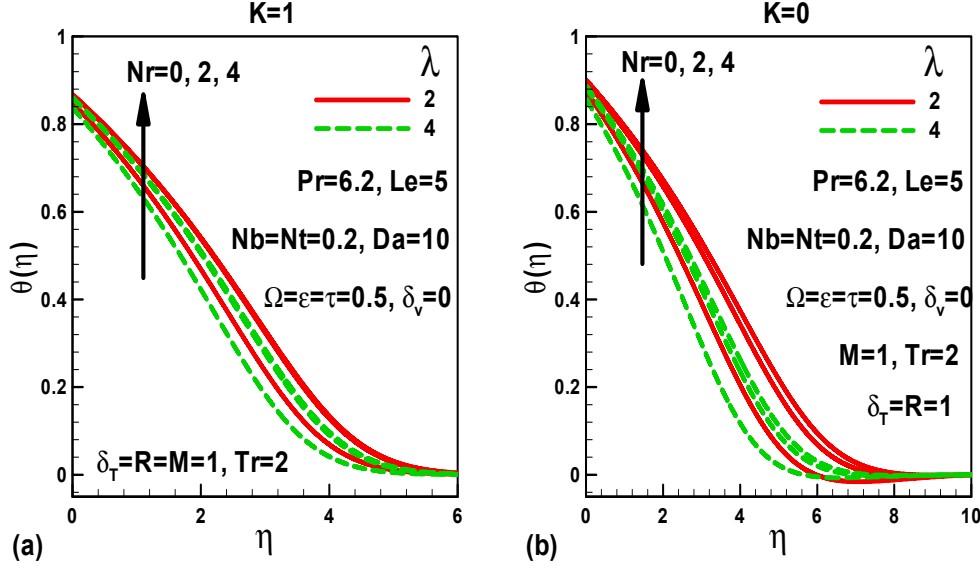

**Figure 5.** Effects of buoyancy ratio and mixed convection parameters on dimensionless temperature (**a**) with pressure gradient and (**b**) without pressure gradient.

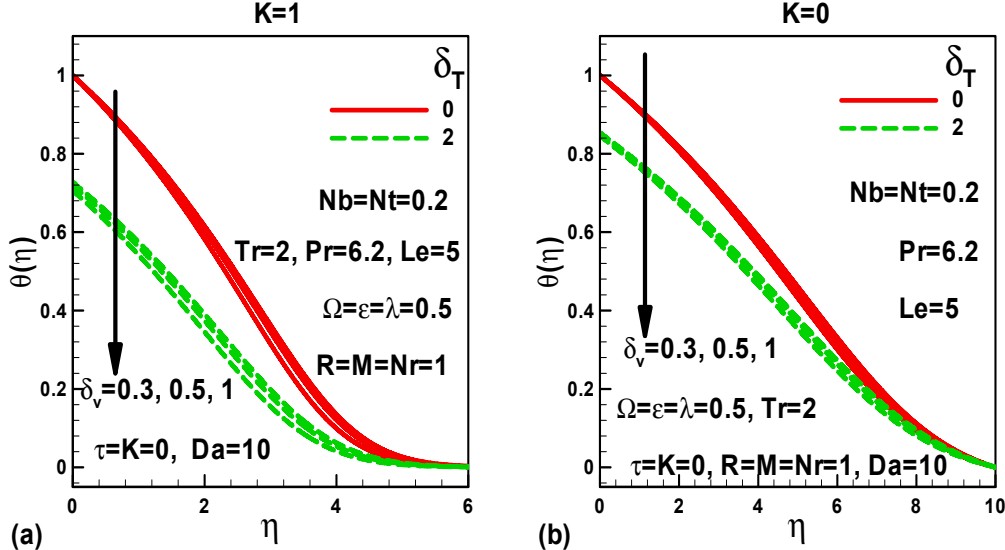

**Figure 6.** Effects of slip parameters on dimensionless temperature (**a**) with pressure gradient and (**b**) without pressure gradient.

Figure 7a,b portray the influences of the permeability parameter, dimensionless time, magnetic field and Roseland conduction-radiation parameters, respectively, on temperature distributions. Temperature is enhanced as the dimensionless time increases (Figure 7a). The impact of Darcy number on temperature is insignificant. Figure 7b indicates that *decreasing* thermal radiation (higher *R* values) reduces temperature for both hydrodynamic ($M = 0$) and magnetohydrodynamic flow ($M \neq 0$). A comparable trend of temperature response was also reported by Hossain et al. [51], which provides further corroboration of the physical accuracy of the present simulated results. Increasing the magnetic field yields a reduction in the temperature, i.e., thinner thermal boundary layer, which again is contrary

to the conventional flat plate hydromagnetic boundary layer flow in a constant free stream (wherein kinetic energy is dissipated as heat in the boundary layer inducing a thicker thermal boundary layer). Again, this deviation is due to the free stream effect in the present model.

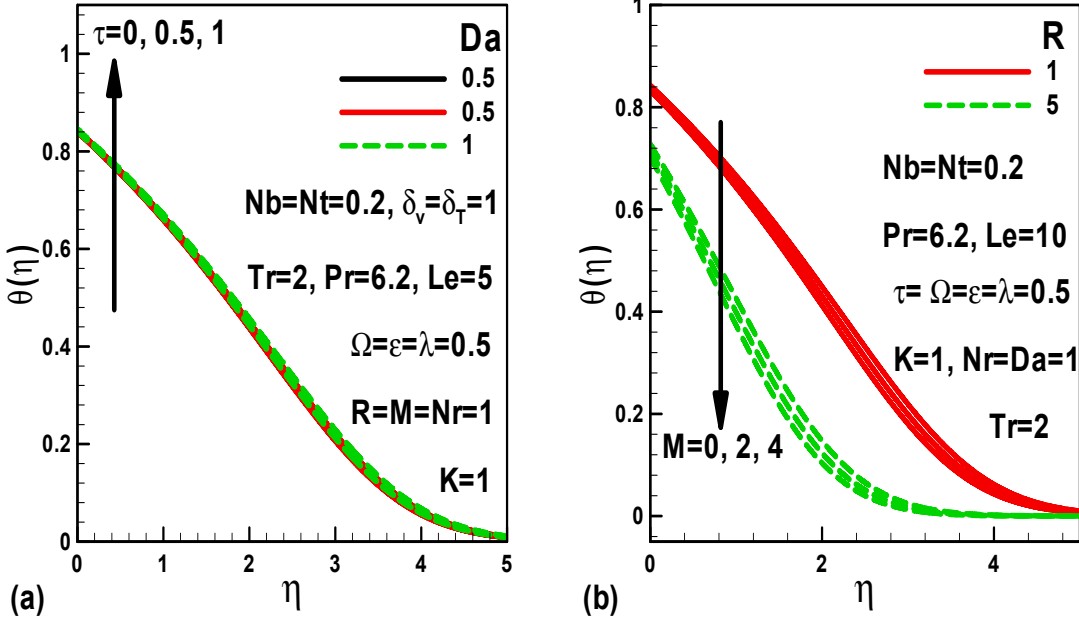

**Figure 7.** Effects of several parameters on dimensionless temperature with pressure gradient.

The impact of the buoyancy ratio ($Nr$), mixed convection and pressure force parameters ($K$) on the dimensionless nanoparticle concentration profiles is illustrated in Figure 8a,b. Figure 8a shows that the dimensionless concentration reduces with buoyancy ratio both in the presence and absence of pressure force. The mixed convection parameter mobilizes a rise in the concentration both with and without buoyancy forces. The higher buoyancy ratio, therefore, decreases concentration boundary layer thickness, whereas, higher mixed convection parameter enhances it.

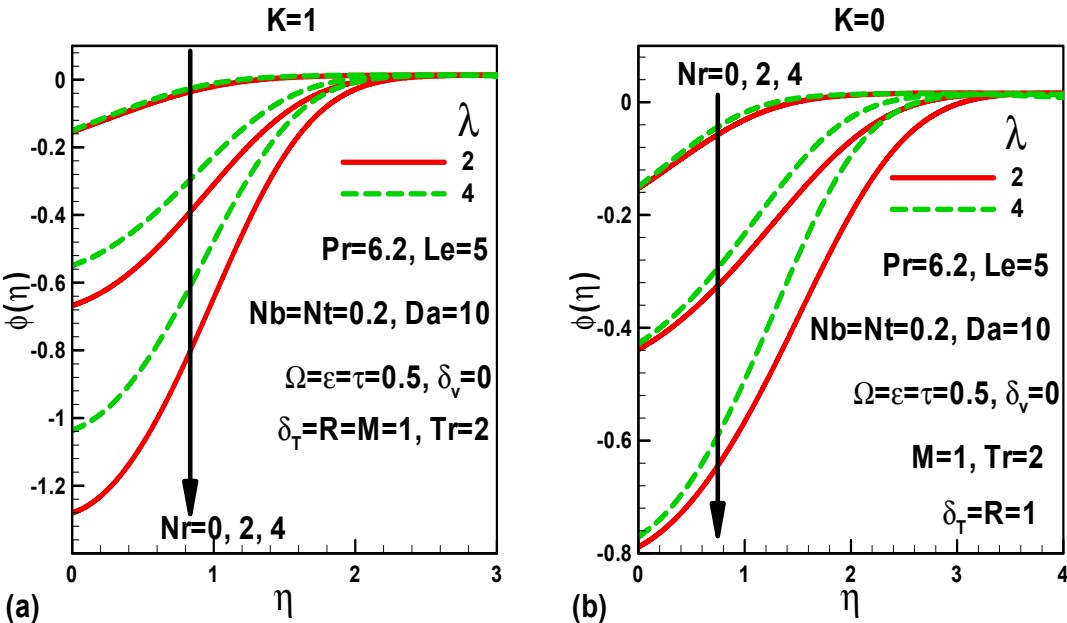

**Figure 8.** Effects of buoyancy ratio and mixed convection parameters on dimensionless concentration (**a**) with pressure gradient and (**b**) without pressure gradient.

Figure 9a,b have been drawn to show the impact of velocity slip and thermal slip on the concentration profiles. Concentration increases as thermal slip parameter increase both in the presence and absence of velocity slip. The velocity slip parameter leads to an enhancement of the concentration for either an isothermal plate or a plate with a thermal slip. All profiles in Figures 1–9 converge asymptotically confirming that a sufficiently large far-field boundary condition is used during the computations. We now appraise the influences of various thermophysical parameters on friction, the heat transfer rates, and mass transfer rates.

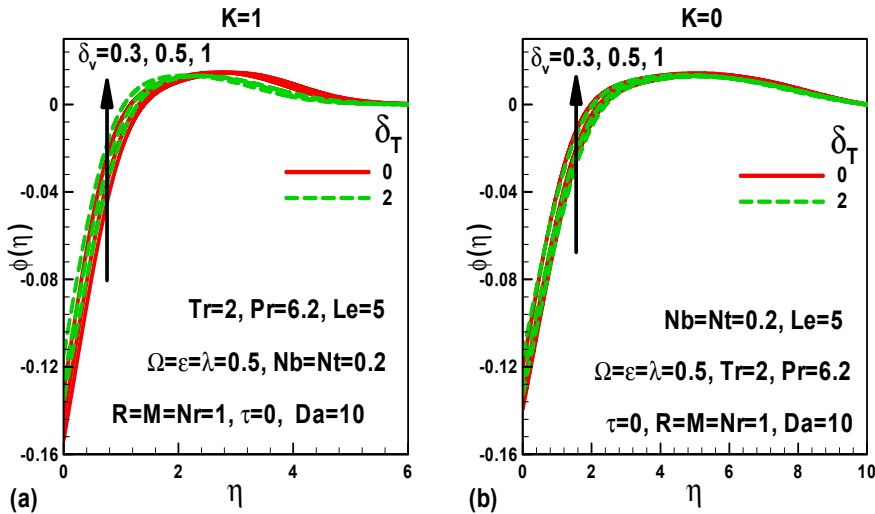

**Figure 9.** Effects of slip parameters on dimensionless velocity (**a**) with pressure gradient and (**b**) without pressure gradient.

Figure 10a,b show the influences of dimensionless time, frequency, and mixed convection parameters on friction factor with/without a pressure gradient, respectively. It is observed that friction is accentuated as mixed convection and frequency parameters increase both with and without a pressure force term. As expected, the dimensionless time reduces friction factor. In other words, with progression in time, the nanoscale boundary layer flow is decelerated, and the momentum boundary layer thickness is increased.

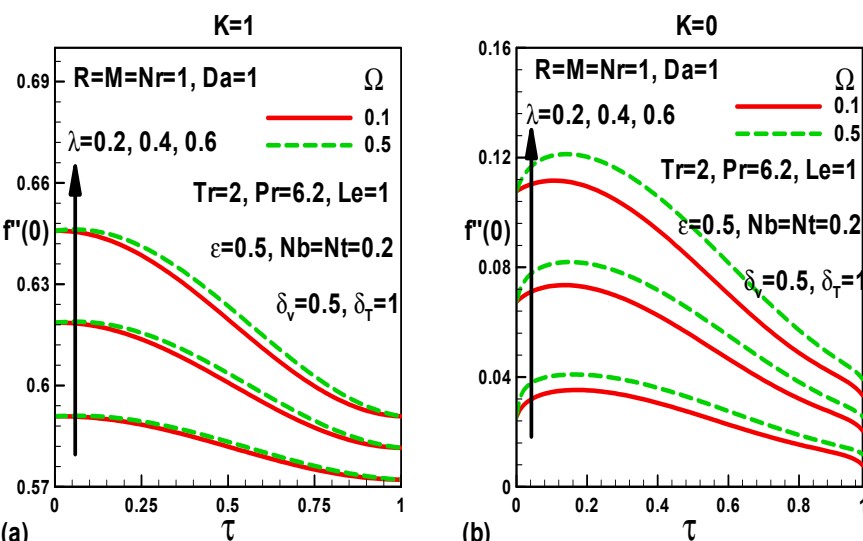

**Figure 10.** Effects of dimensionless time, frequency, and mixed convection parameters on skin friction (**a**) with pressure gradient and (**b**) without pressure gradient.

Figure 11a,b are portrayed to show the effect of Darcy number, velocity slip, and magnetic field parameters on skin friction for a *moving* free stream and a *stationary* free stream, respectively. As expected, slip velocity reduces the friction factor for both a moving and quiescent free stream. This is due to the slope of the velocity at the wall decreasing, which can be perceived from the boundary condition $f'(0) = \delta_v f''(0)$. As $\delta_v \to \infty$ (full slip), the shear stress at the wall is close to zero. Increasing Darcy number decreases friction factor when free stream velocity is moving ($K = 1$), and the opposite trend is noticed for a stagnant free stream ($K = 0$). Elevation in the magnetic field parameter increases the friction when the free stream velocity is moving ($K = 1$), and the opposite trend is observed for a calm, free stream ($K = 0$).

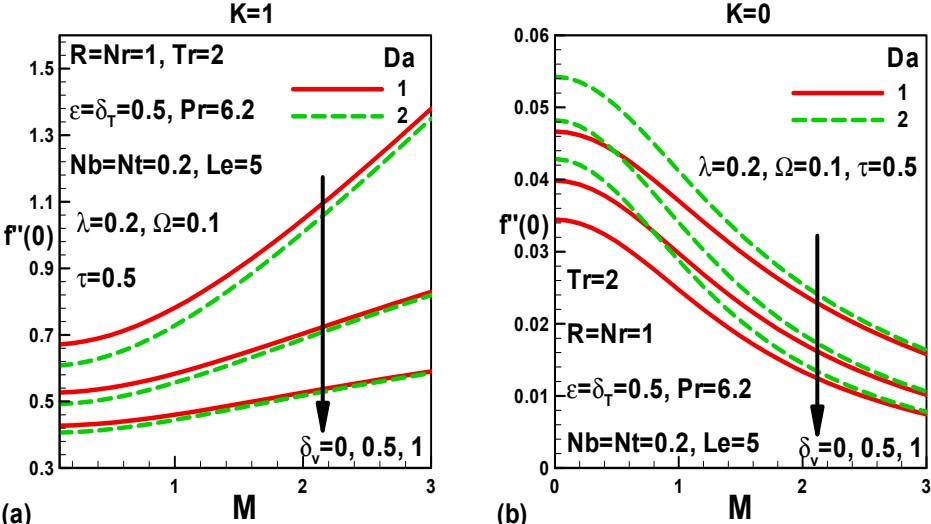

**Figure 11.** Effects of Darcy number, velocity slip, and magnetic parameters on skin friction (**a**) with pressure gradient and (**b**) without pressure gradient.

Figure 12a,b display the influence of time, frequency, and mixed convection parameters on heat transfer rates. The heat transfer rate is elevated as mixed convection, and frequency parameters increase for both the moving free stream and stationary free stream. However, Figure 12b indicates that heat transfer rate is decreased with larger values of the dimensionless time.

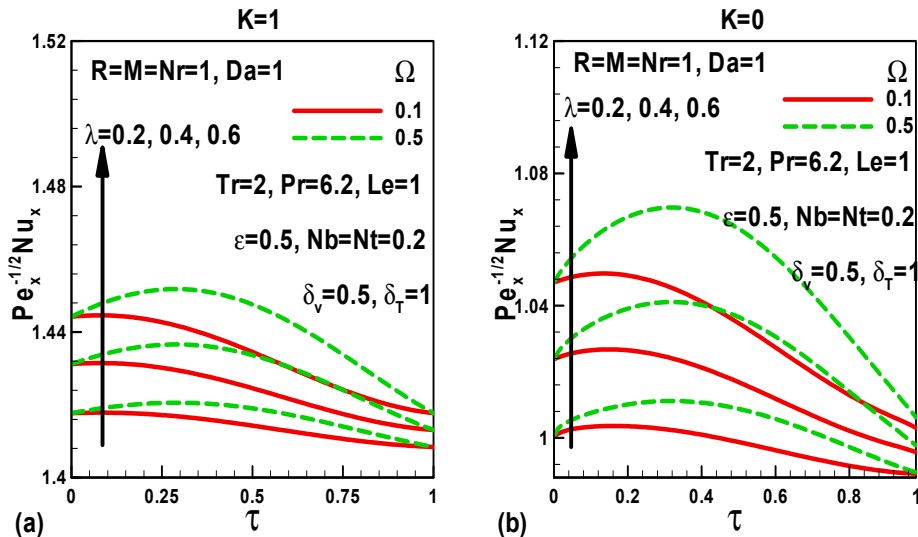

**Figure 12.** Effects of dimensionless time, frequency, and mixed convection parameters on Nusselt numbers (**a**) with pressure gradient and (**b**) without pressure gradient.

Figure 13a,b elucidate the impact of Darcy number, velocity slip, and magnetic parameters on Nusselt numbers for the case of a moving free stream (Figure 13a) and quiescent free stream (Figure 13b). Darcy number increases (decreases) the heat transfer rates in the case of a stationary (moving) free stream. Velocity slip leads to an escalation in heat transfer rates for both moving and stationary free streams. Magnetic field decreases (increases) heat transfer rates for the case of a stationary (moving) free stream.

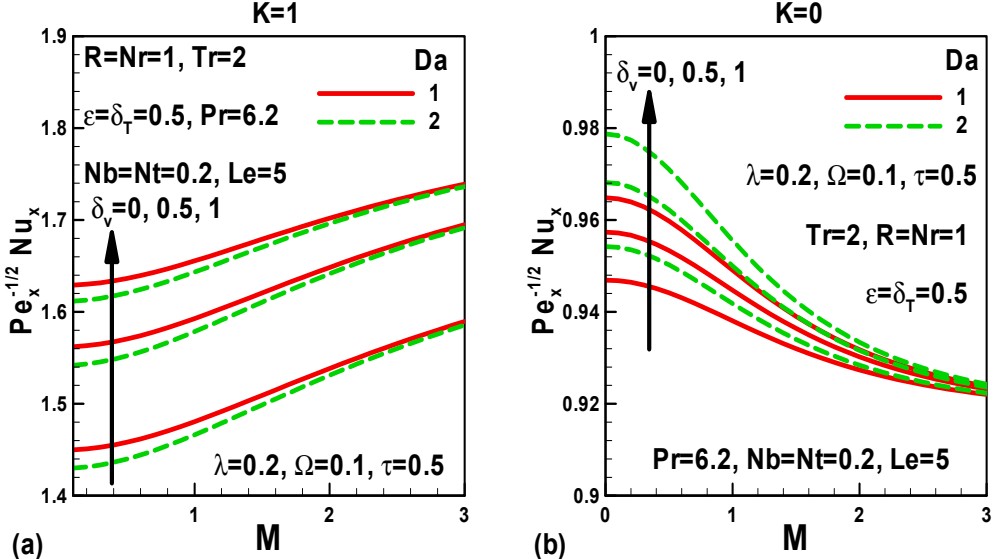

**Figure 13.** Effects of Darcy number, velocity slip, and magnetic parameters on Nusselt numbers (**a**) with pressure gradient and (**b**) without pressure gradient.

Figure 14a,b provide the response of heat transfer rates corresponding to the thermal slip, conduction-radiation and thermophoresis parameters. Heat transfer rates are suppressed with greater values of conduction-radiation parameter, thermal slip parameter and thermophoresis parameter for both moving and free stream velocity. Increasing thermal radiative heat transfer, therefore, reduces heat transfer rates, since an increase in thermal energy migrating to the nanofluid is generated (which implies a reduction in heat transfer to the plate), manifesting in thicker thermal boundary layers in the regime.

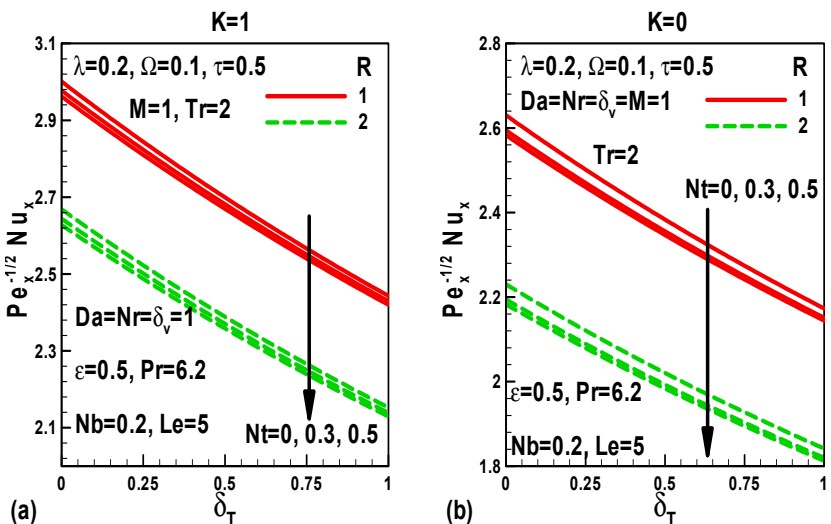

**Figure 14.** Effects of thermal slip, radiation, and thermophoresis parameters on Nusselt numbers (**a**) with pressure gradient and (**b**) without pressure gradient.

## 6. Conclusions

A mathematical model has been presented for g-jitter induced mixed convective transient two-dimensional viscous MHD flow of nanofluid past a nonlinearly radiating plate in a Darcian porous medium with multiple wall slip boundary conditions. The governing equations were rendered into the non-similar form using coordinate transformations and multiple slips and passively controlled boundary conditions deployed to achieve more physically realistic and practically applicable results. A finite-difference numerical method with Bellman-Kalaba quasilinearization was used to solve the transformed nonlinear two-parameter boundary value problem. The main findings of the present investigation are as follows:

(i)     In the case of stationary free stream, surface friction rises with Darcy number, whereas, it reduces with magnetic field and time.

(ii)    In the case of moving free stream, friction decreases with Darcy number and time; however, it increases with the magnetic field.

(iii)   Increasing velocity slip reduces friction factor, whereas, the mixed convection and frequency parameters lead to an increase in friction for both moving and stationary free streams.

(iv)   Heat transfer rates rise with greater values of mixed convection, velocity slip, and frequency; conversely, they reduce with greater thermophoresis, thermal slip, and conduction-radiation (i.e., lower radiative flux) parameters for both moving and stationary free streams.

(v)    In the case of moving (stationary) free stream, the heat transfer rate decreases (increases) with an increase in the Darcy number.

(vi)   In the case of moving (stationary) free stream, the heat transfer rate increases (decreases) with greater magnetic field parameter.

The present computations have addressed Newtonian nanofluids. Future studies will consider rheological effects and will be communicated soon.

**Author Contributions:** M.J.U. Formulation and writing abstract and conclusions. W.A.K. Solved the problem numerically and write Results and Discussion Section. O.A.B. Introduction, Literature review and arranging references. A.I.M.I. Overall supervision and funding. Review the manuscript. All authors have read and agreed to the published version of the manuscript'

**Funding:** This research was funded by Universiti Sains Malaysia, grant number 1001/PMATHS/8011013" and "The APC was funded by Universiti Sains Malaysia.

**Conflicts of Interest:** The authors declare no conflicts of interest.

## Nomenclature

| | |
|---|---|
| $a$ | constant (1/s) |
| $B_0$ | strength of magnetic field (Wb/m$^2$) |
| $c_p$ | specific heat at constant pressure (kJ/kg-K) |
| $C$ | nanoparticle volume fraction (-) |
| $C_{fx}$ | friction factor (-) |
| $C_w$ | wall nanoparticle volume fraction (-) |
| $C_\infty$ | Surrounding nanoparticle volume fraction (-) |
| $D_B$ | Brownian diffusion coefficient (m$^2$/s) |
| $D_T$ | thermophoretic diffusion coefficient (m$^2$/s) |
| $Da$ | Darcy number (-) |
| $f$ | stream function (-) |
| $\mathbf{g}^*(t)$ | the gravitational acceleration (m/s$^2$) |
| $g_0$ | time-averaged value of the gravitational acceleration $\mathbf{g}^*(t)$ (m/s$^2$) |
| $k$ | thermal conductivity (W/m-K) |
| $k_p$ | permeability (m$^2$) |
| $\overrightarrow{K_1}$ | unit vector |

| $K$ | constant reflecting pressure gradient term (-) |
|---|---|
| $L$ | characteristic length (m) |
| $Le$ | Lewis number (-) |
| $M$ | magnetic field parameter (-) |
| $Nb$ | Brownian motion parameter (-) |
| $Nr$ | buoyancy ratio parameter (-) |
| $Nt$ | thermophoresis parameter (-) |
| $Nu_x$ | local Nusselt number (-) |
| Pr | Prandtl number (-) |
| $q_m$ | wall mass flux (m/s) |
| $q_r$ | radiative heat flux (J/m$^2$-s) |
| $q_w$ | wall heat flux (W/m$^2$) |
| R | Radiation-conduction parameter (-) |
| Re | Reynolds number (-) |
| $Re_x$ | local Reynolds number (-) |
| $Sh_x$ | local Sherwood number (-) |
| $t$ | time (s) |
| $T$ | temperature (K) |
| $Tr$ | temperature ratio parameter (K) |
| $T_w$ | sheet temperature (K) |
| $T_\infty$ | ambient temperature (K) |
| $u$ | velocity component along the $x$-axis (m/s) |
| $v$ | velocity component along the $y$-axis (m/s) |
| $u_e$ | external velocity (m/s) |
| $x, y$ | Rectangular coordinates lengthwise and perpendicular to the plate (m) |

Greek symbols

| $\alpha$ | thermal diffusivity of the fluid (m$^2$/s) |
|---|---|
| $\beta_T$ | thermal expansion coefficient (1/K) |
| $\beta_C$ | coefficient of mass expansion (-) |
| $\delta_v$ | velocity slip parameter (-) |
| $\delta_T$ | thermal slip parameter (-) |
| $\omega$ | frequency of oscillation (1/s) |
| $\Omega$ | non-dimensional frequency (-) |
| $\lambda$ | mixed convection parameter (-) |
| $\eta$ | similarity independent variable (-) |
| $\theta$ | non-dimensional temperature (-) |
| $\phi$ | nanoparticle volume fraction (-) |
| $\mu$ | dynamic viscosity of the fluid (Ns/m$^2$) |
| $v$ | kinematic viscosity of the fluid (m$^2$/s) |
| $\rho$ | density of the base fluid (kg/m$^3$) |
| $(\rho c)_f$ | volumetric heat capacity of the fluid (J/m$^3$-K) |
| $(\rho c)_p$ | volumetric heat capacity of the nanoparticle material (J/m$^3$-K) |
| $\sigma$ | electric conductivity (Siemens/m) |
| $\sigma_1$ | Stefan-Boltzmann constant (W/m$^2$-K$^4$) |
| $\tau_1$ | ratio of the effective heat capacity of the nanoparticle to the fluid heat capacity (-) |
| $\psi$ | stream function (m$^2$/s) |

Subscripts

| $()'$ | denotes ordinary differentiation with respect to $\eta$ |
|---|---|

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
