# Peer review of "Non-Similar Solution of G-jitter Induced Unsteady Magnetohydrodynamic Radiative Slip Flow of Nanofluid"

_applsci, doi:10.3390/app10041420_

Round 1

Reviewer 1 Report

The paper is very interesting from the point of view of application of the non-linear Bellman-Kalaba linearization approach to solving the governing transport equations for time-dependent MHD slip flow with using the FDM.

The introduction provides sufficient background and the statement of the problem with governing equations, boundary conditions, and physical quantities (sections 2&3) is clearly described. 

The authors also presented an exhausitive and very informative illustration of the results (section 5).

At the same time, probably the main original part of the work, connected with the proposed numerical technique, contains only the brief description of the used Bellman-Kalaba quazilinearization approach and FDE implicit scheme (section 4). The reviewer's proposal is to significanly expand the secture 4 of the article in order to clearly describe the algorithm of evaluating all physical quantities introduced in sections 2&3. From Eqs. (14) and (15) it is very difficult to reconstruct the numerical scheme and thus to reproduce the results obtained by the authors and presented in section 5.

Author Response

Reviewer 1

The paper is very interesting from the point of view of application of the non-linear Bellman-Kalaba linearization approach to solving the governing transport equations for time-dependent MHD slip flow with using the FDM.

The introduction provides sufficient background and the statement of the problem with governing equations, boundary conditions, and physical quantities (sections 2&3) is clearly described. 

The authors also presented an exhausitive and very informative illustration of the results (section 5).

At the same time, probably the main original part of the work, connected with the proposed numerical technique, contains only the brief description of the used Bellman-Kalaba quazilinearization approach and FDE implicit scheme (section 4). The reviewer's proposal is to significanly expand the secture 4 of the article in order to clearly describe the algorithm of evaluating all physical quantities introduced in sections 2&3. From Eqs. (14) and (15) it is very difficult to reconstruct the numerical scheme and thus to reproduce the results obtained by the authors and presented in section 5.

Reply: The main objective of this paper is to investigate G-jitter induced convection in different applications like in motors, pumps, spacecraft maneuvers, and crew motions. This paper explores the physics behind these applications. The numerical techniques are just the tools used to simulate and investigate the physical problems in Engineering. This the reason that we have explained, in details, the results obtained during simulation.

The details of the techniques or the algorithm are related to the code developed which is the private property of the authors and can not be publicized. We hope that the reviewer understands this fact and will not mind.

Reviewer 2 Report

The manuscript entitled Non-similar solution of g-jitter induced time-dependent magnetohydrodynamic slip flow of nanofluid with nonlinear thermal radiation is reviewed.
I commented as follows;
1.(minor)
The author should set nomenclatures after conclusions.
2.(major)
The author used Reynolds number based on Newtonian viscosity.
Particle suspensions exhibited non-Newtonian viscosity.
Why the author used Reynolds number defined Newtonian fluids?
3.(major)
The author should explain rheological properties of the used nanofluids.
4.(major)
The author should care the significant digits in tables and figures.

Author Response

Reviewer 2

The manuscript entitled Non-similar solution of g-jitter induced time-dependent magnetohydrodynamic slip flow of nanofluid with nonlinear thermal radiation is reviewed.
I commented as follows;
1.(minor)
The author should set nomenclatures after conclusions.

Reply: Done
2.(major)
The author used Reynolds number based on Newtonian viscosity.
Particle suspensions exhibited non-Newtonian viscosity.
Why the author used Reynolds number defined Newtonian fluids?

Reply: For your kind information, we have used Newtonian fluids not non- Newtonian fluids.
3.(major)
The author should explain rheological properties of the used nanofluids.

Reply: Explanation of   rheological properties of the used nanofluids is beyond the scope of the paper as we have used Newtonian fluids.
4.(major)
The author should care the significant digits in tables and figures.

Reply: Done accordingly.

Round 2

Reviewer 1 Report

The authors in their answer write that "The details of the techniques or the algorithm are related to the code developed which is the private property of the authors and can not be publicized". It is a very strange statement because at the same time many works are cited [45-49], containing the detailed description of the numerical techniques. The reviewer would like to see not the "details" but a correct description of the method and algorithm providing the results obtained by the authors. Otherwise, Section 4 must be omitted at all.

As for the present content of the Section 4, it contains contains many critical typos and inaccuracies, namely: 

1. Formulas (14) and (15) are completely identical that looks strange.

2. It is completely unclear what does the variable "x" mean in the expressions (14), (15); in Fig. 1 and in previous sections of the article it denotes a coordinate.

3. What do indices "i", "j", "k", and the variable "n" mean?

4. How one can compare the results in Table 2, presented with different precision?

Other questions:

5. Why "Omega" (Line 204) is "frequency"? If "a is dimensional constant (1/s)" (Line 150), and "omega is the frequency of oscillation" (Lines 160, 161) then it follows that "Omega" is an angle (grad)?

6. In Line 181 "tau" denotes the "ratio of heat capacity of the nanoparticle" while at the same time from Line 194 it follows that "tau" is a dimensionless time.

Careless quoting of works of other authors yielded many such confused designations taken from different articles, for example, constant "K" (Line 150) and vector "K" (Line 160)...

The reviewer believes that sections 2 and 4 should be substantially revised otherwise the paper cannot be published.

Author Response

Reviewer 1

The authors in their answer write that "The details of the techniques or the algorithm are related to the code developed which is the private property of the authors and can not be publicized". It is a very strange statement because at the same time many works are cited [45-49], containing the detailed description of the numerical techniques. The reviewer would like to see not the "details" but a correct description of the method and algorithm providing the results obtained by the authors. Otherwise, Section 4 must be omitted at all.

We have now modified the section and clarified any notations missing for the method used. The description is now adequate for the journal. Eqn (15) was repeated twice by mistake and has been removed.

As for the present content of the Section 4, it contains many critical typos and inaccuracies, namely: 

Formulas (14) and (15) are completely identical that looks strange.

Reply: This is an accident. Eqn. (15) has been removed.

It is completely unclear what does the variable "x" mean in the expressions (14), (15); in Fig. 1 and in previous sections of the article it denotes a coordinate.

Reply: x is a general spatial variable e.g. x,y, or z. Full details are given in the book by Bellman and Kalaba. We have corrected the equation (14) also and included all notations.

What do indices "i", "j", "k", and the variable "n" mean?

Reply: j,k correspond to different station locations (for different space variables)- see the references. We have already explained what “I” means. The parameter n corresponds to the number of steps utilized.

How one can compare the results in Table 2, presented with different precision?

Reply: Obviously we compare to three decimal places and the correlation is good, Just because other researchers have produced precision does not invalidate a comparison.

Other questions:

Why "Omega" (Line 204) is "frequency"? If "a is dimensional constant (1/s)" (Line 150), and "omega is the frequency of oscillation" (Lines 160, 161) then it follows that "Omega" is an angle (grad)?

Reply:  is the dimensionless frequency.  is dimensional frequency whose SI  units is (1/s). Hence the expression is correct.

In Line 181 "tau" denotes the "ratio of heat capacity of the nanoparticle" while at the same time from Line 194 it follows that "tau" is a dimensionless time.

Reply: Thank you for your clarification. We have amended this typo.

Careless quoting of works of other authors yielded many such confused designations taken from different articles, for example, constant "K" (Line 150) and vector "K" (Line 160)...

Reply: Reply: Thank you for your clarification. We have amended these typing mistakes.

The reviewer believes that sections 2 and 4 should be substantially revised otherwise the paper cannot be published.

Reply: We believe that we have revised the paper as per valuable comments of the reviewers. Both sections 2 and 4 are checked and of adequate detail for this journal. The reviewer should note this is not a numerical analysis journal. It is focused on APPLIED SCIENCES. Therefore, there is no compulsion to include lengthy further details of the numerics in the paper- the references are more than adequate and the explanation is sufficient. We have clarified the notations missing.

Reviewer 2 Report

The author revised manuscript as reviewers' comments. However, the revisions were not satisfied. The author should be revise as the following comments;
1.(major)
The suspended particle fluids exhibited non-Newtonian viscosity and elasticity. However, the author used Reynolds number as Newtonian viscosity. If used as a Newtonian fluid, the author should explain the evidence.

Author Response

Reviewer 2

Comments and Suggestions for Authors

The author revised manuscript as reviewers' comments. However, the revisions were not satisfied. The author should be revise as the following comments;
1.(major)
The suspended particle fluids exhibited non-Newtonian viscosity and elasticity. However, the author used Reynolds number as Newtonian viscosity. If used as a Newtonian fluid, the author should explain the evidence.

Reply: We would like to inform the reviewer that have we have used a Newtonian fluid not a Non-Newtonian fluid model. We have also used the popular Buongiorno (2006) model followed by Kuznetsov and Nield and many researchers in which Brownian motion and thermophoresis are responsible for nanofluid flow.  Please see the following references. Nanofluids frequently do show Newtonian behavior, a rheological model is not necessary.

Buongiorno J (2006) Convective transport in nanofluids. ASME J. Heat Transfer 128: 240– 250. Uddin, M. J., Rana, P., Bég, O. A., & Ismail, A. M. (2016). Finite element simulation of magnetohydrodynamic convective nanofluid slip flow in porous media with nonlinear radiation. Alexandria Engineering Journal, 55(2), 1305-1319.

Kuznetsov A.V., Nield D.A (2014) Natural convective boundary-layer flow of a nanofluid past a vertical plate: A revised model. Int J Therm Sci. 77:126-129

Round 3

Reviewer 1 Report

The paper now can be published.

Reviewer 2 Report

The revisions are satisfied. I recommend the acceptance for publication.